# Effect of Heat Treatment on the Microstructure and Properties of Ti600/TC18 Joints by Inertia Friction Welding

**DOI:** 10.3390/ma16010392

**Published:** 2022-12-31

**Authors:** Yingying Liu, Kaixin Ren, Wantao Tian, Xiaolong Shangguan, Siyu Tan, Qihao Yang

**Affiliations:** School of Metallurgical Engineering, Xi’an University of Architecture and Technology, Xi’an 710055, China

**Keywords:** inertia friction welding, microstructure, properties, stress-relief annealing, Ti600/TC18, two-stage annealing

## Abstract

The Ti600/TC18 dissimilar titanium alloy joints were prepared by inertia friction welding (IFW). Then, stress-relief annealing and two-stage annealing were performed to optimize the microstructure and properties of the original joints, the purpose of them is to improve the structure and performance of the joints. Then, the microstructure, phase composition, tensile properties, microhardness, and fracture morphology of the joints after heat treatments were investigated. The results showed that after stress-relief annealing, the microstructure of the joints was almost similar to that of the specimen before annealing; the weld zone (WZ) of the joints was composed of fine recrystallized grains and α′, and the more β phases underwent a martensitic transformation. The shapes and sizes of α_p_ phases were increased after two-stage annealing; its percentage content was decreased. The tensile properties and the microhardness values of the joints undergoing stress-relief annealing were relatively higher than that of the joints undergoing two-stage annealing; there was no obvious change in the plasticity of the joints. It was confirmed that the stress-relief annealing microstructure was composed of α′ and β phases, which were beneficial to the properties of the joints. However, the α_s_ phases were coarsened after two-stage annealing, and the properties of the joints were reduced.

## 1. Introduction

Titanium alloys have been widely applied to crucial components in aero engines due to their perfect combination of properties, such as low density, excellent corrosion resistance, and the high strength-to-weight ratio [1,2]. Ti600 alloy (Ti-6Al-2.8Sn-4Zr-0.5Mo-0.4Si-0.1Y) is a near-α alloy with good comprehensive properties, especially excellent creep properties. It is used for a long time at 600 °C and can be used in compressor impellers [3]. The TC18 alloy (Ti-5Al-5Mo-5V-1Cr-1Fe) is an α + β alloy with high strength and toughness. It is often used to prepare the main load-bearing structural parts of the aero engine [4].

There are many outstanding advantages in welding technology, such as simplifying mechanical mechanisms, which can reduce production costs and improve performance. Among the numerous welding methods, IFW, as an efficient and advanced joining method, has been widely used to join dissimilar alloy parts. It can effectively avoid defects such as holes and cracks due to metal melting. IFW has become the key technology in the manufacture and repair of Ti-alloy integral discs in aero-engines. [5] While Ti600 high-temperature alloys and TC18 high-strength alloys are connected to form dual-alloy compressor discs, they can be used in the rear stages of the compressor blade disc. Furthermore, they can meet performance requirements and reduce manufacturing costs [6,7].

For welded structural parts, defects such as residual stresses tend to appear in the WZ and heat-affected zone (HAZ), which can lead to joint deformation or cracking [8,9].

A large number of studies have shown that changing the welding process and parameters [10], adding appropriate post-weld treatment or deformation and other methods can improve the overall performance of the welding interface [11,12].

Mehmet et al. [10,13] optimized the MIG welding groove using the Taguchi method. The results showed that joints with different performance requirements can be obtained using different welding-process parameters. Cheng et al. [14] showed that the tensile strength of Ti55 electron-beam-welded joints was improved after annealing at 500 °C. Zhang et al. [15] showed that the thickness of the IMC layer of the 6082 aluminum/TC4 alloy friction stir welding interface was increased with increasing annealing temperature or holding time, and the shear strength of the joint after heat treatment was increased. Wang et al. [16] adjusted the flow-stress difference between the weld seam of the laser-welded titanium alloy pipe and the base metal by 950 °C/2 h, air cooling + 600 °C/2 h, and air cooling. The results showed that the maximum flow-stress difference ratio decreased from 36% to 17% after annealing, and the deformation uniformity of the expanded pipe increased by 24.6%. Gao et al. [17] compared the effect of primary annealing and two-stage annealing on the performance of TC4-DT/TC21 linear-friction-welded head law. The results showed that two-stage annealing was significantly better than prior annealing, strength and the impact energy were increased by 91 MPa and 24.4 J, and the plasticity was still better.

In order to improve the microstructure and properties of the Ti600/TC18 IFW joints, stress-relief annealing and two-stage annealing are carried out. The microstructure and properties of joints after different heat treatments are investigated, respectively. Moreover, the relationship between welding-process parameters, microstructure, and properties is obtained. Finally, the technology and parameters are provided for preparing Ti600/TC18 IFW joints with good interface performance. Therefore, the results can provide a theoretical basis and technical support for researching dual titanium alloys.

## 2. Materials and Methods

### 2.1. Materials

The main chemical composition and mechanical properties of Ti600 and TC18 are shown in Table 1 and Table 2. Figure 1 shows the microstructures of two alloys. Ti600 alloy consisted of equiaxed primary α phases and a small amount of lamellar α + β phases with clear phase boundary (Figure 1a); its phase-transition temperature is 1010~1015 °C. TC18 alloy is composed of needle-like α phases and is composed of net basket microstructure (Figure 1b); its phase-transition temperature is about 875 °C.

### 2.2. Experimental Processs

The Ti600/TC18 dissimilar titanium alloy joints were welded by the IFW machine (250BX, Newburyport, MA, USA). The sample sizes of the Ti600 alloy ring and TC18 alloy inner shaft were Φ 107 mm × 60 mm and Φ 73.5 mm × 60 mm, respectively. They were cleaned integrally with acetone solution; the purpose was to remove impurities such as rust, residual oxide, moisture, and residual lubricating oil from their surfaces before welding. Then, the welding was carried out with the moment of rotational inertia being 105.34 kg·m^2^, the upsetting pressure 5512 KPa, and the rotating speed 600 r/min.

The vacuum heat treatment tube furnace (GSL-1700X, Hefei, China) was used to perform stress-relief annealing and two-stage annealing for the Ti600/TC18 joints. The heat-treatment process parameters are shown in Table 3 and Table 4, respectively. The metallographic and tensile samples were prepared by the disc-shaped Ti600/TC18 joint, as shown in Figure 2.

The Ti600/TC18 metallographic samples after heat treatment were polished with sandpaper of 240#, 400#, 600#, 1000#, 1500#, 2000#, and 3000#, respectively. Therefore, there was no obvious scratch on their surface. Then, it was polished with a silica suspension until its surface was mirrored; finally, the polished sample was ultrasonically cleaned. The solution used for ultrasonic cleaning was deionized water and anhydrous ethanol.

The microstructure of the joints and its fracture morphology were observed by scanning electron microscope (SEM, VEGA II TESCAN, Brno, The Czech Republic), and the ratio of corrosion solution was HF:HNO_3_:H_2_O = 2:10:88. The phase composition of the joints was determined by the X-ray diffractometer (XRD, D8 ADVANCE A25, Karlsruhe, Baden-Württemberg, Germany).

The size of the tensile samples is shown in Figure 3. Tensile properties tests were undergone on the electronic universal testing machine (Instron 5985, Boston, MA, USA); the room temperature and high temperature (450 °C) tensile tests were conducted by GB/T 228-2010 and GB/T 4338-2006, respectively.

The microhardness values of joints were tested by a micro-Vickers hardness tester (BUEHLER MICROMET5104, Akashi Corporation, Osaka, Japan), where the test load was 0.98 N and the loading time was 10 s. The microhardness values of the joints were measured from the starting point, which was the intersection point of the WZ and HAZ of the Ti600 alloy. A point was measured at an interval of 500 μm from the origin to both sides. Finally, the points reached 2250 μm to the left of the starting point; they reached 3750 μm to the right of the starting point.

## 3. Results and Discussion

### 3.1. Microstructure of Ti600/TC18 IFW Joints after Heat Treatment

The microstructure of the Ti600/TC18 joints before heat treatment is shown in Figure 4. It can be seen that the microstructure of the joints is composed of three distinct regions: the WZ, the HAZ of Ti600 alloy, and the HAZ of TC18 alloy. The microstructure of the joints in the WZ exhibits a dynamic recrystallization microstructure. The α phases and elongated needle-like α′ phases are observed in the HAZ of Ti600 alloy, while a combination of needle-like α phases and β phases are observed in the HAZ of TC18 alloy.

Due to the plastic deformation, dynamic recrystallization occurs in the WZ, so the refined dynamic recrystallization grains are formed. In addition, the microstructures of the joints on both sides are elongated under the influence of welding pressure. Moreover, the temperature of the WZ is higher than the transition temperature of the β phase during welding, so the original α phases were transformed into the β phases. After welding, with the rapid cooling of the WZ temperature, it is too late to precipitate a large number of α phases. In addition, in the metastable β phase, the needle-like α phases are grown along the direction of the grain boundary. The martensite transformation occurred, and the needle-like α′ phases are generated [18,19].

Figure 5 shows the microstructure of the Ti600/TC18 joints after stress-relief annealing. It can be seen that after stress-relief annealing, the equiaxed α, secondary lamellar α′ phases, and residual β phases are visible in the WZ. The HAZ of Ti600 alloy consists of deformed primary α phases, secondary lamellar α′ phases, and β phases; the equiaxed crystals are formed on the metastable β matrix. The lamellar α′ phases and β phases are characteristic of the HAZ of TC18 alloy. Moreover, as the temperature is increased, the needle-like α phases in the HAZ of Ti600 alloy and TC18 alloy are both gradually grown and coarsened; more β phases undergo a martensitic transformation. The microstructure of the joints is more uniform (Figure 5c).

Compared with Figure 4, the microstructure of the joints is almost similar to that of the specimen before annealing. The lamellar α′ phases are coarsened, and the grain boundary of α phases are discontinuous in Figure 5. This is related to the fact that the metastable β grains undergo an aging decomposition after stress-relief annealing. A large number of secondary α and β phases are precipitated, which can improve the strength of the joints [8,16].

Figure 6 shows the microstructure of the Ti600/TC18 welded joints after two-stage annealing. There are many refined grains and needle-like martensite α′ phases in the WZ. The α phases and martensite α′ phases dispersed on the β matrix are observed in the HAZ of Ti600 alloy, and the microstructure of the TC18 side consists of α_p_ phases and α_s_ phases with different shapes and sizes.

The phase-transition temperature of Ti600 alloy is higher than that of two-stage annealing. Therefore, the transformation of α → α + β on the Ti600 alloy side is not completed. The microstructure of the Ti600 alloy side is less affected by the temperature of each stage of two-stage annealing. It is observed that after annealing, the microstructure on the Ti600 side consists of equiaxed grains. With the increase in annealing temperature, the size of the grains is larger and more uniform. This is because the increase in the annealing temperature facilitates the growth of recrystallized grains [20].

The purpose of annealing at high temperature is to eliminate the influence of deformation by static recrystallization while retaining a certain amount of metastable phase. On the TC18 alloy side, with the temperature of the high-temperature stage increasing (II-1, II-2, II-4), the grain size of α_p_ phases is increased, and its percentage content is slightly decreased. In addition, the α_s_ phases are precipitated from the transformed β matrix and grown progressively [14]. In Figure 6a, there are many fine α_p_ phases distributed on the transformed β matrix, and a_s_ phases begin to precipitate diffusely at 840 °C. In Figure 6b, the grain size of α_p_ phases is increased, and its percentage content slowly decreases at 860 °C. Moreover, the α_s_ phases are coarsened and have a specific directionality. In Figure 6c, the α_p_ phases are almost disappeared at 880 °C; the net-basket microstructure is characteristic of the long and straight lamellar α_s_ phases, which have a particular directionality.

The α_p_ is formed during high-temperature annealing in the two-phase region and remained in the subsequent air-cooling process. α_s_ is obtained by the transformation of β phases, which are formed during high-temperature annealing and remained in the subsequent air-cooling process. With the increase in high-temperature annealing temperature, the content of α_p_ is decreased, and the more secondary lamellar α phase transforms from the β phase.

The purpose of annealing at low temperatures is to promote the decomposition of the metastable phase in a certain way, which is produced during high-temperature annealing, thus causing aging strengthening. There is no significant variation in the α_p_ phases with the temperature increase of the low-temperature stage (II-3, II-4); while the deformed structure α_s_ phases are gradually spheroid, the grain size of α_s_ phases and their content are both increased. The α_s_ phases are directionality distributed on the transformed β matrix at 590 °C (Figure 6c); they are coarsened with an obvious direction at 610 °C (Figure 6d).

Generally speaking, in the low-temperature annealing process, the lower the annealing temperature, the finer the decomposition products, and the greater the dispersion of the α_s_ precipitated from the matrix.

In summary, the ratio of α_p_ phases and α_s_ phases is influenced by the high-temperature stage; the slip is hindered by the interface of the striped α_s_ phases, which causes the Ti600/TC18 joints to have higher strength and lower plasticity. The precipitation methods of the metastable β are influenced by the low-temperature stage, and the strength of the joints is also affected.

Comparing Figure 5 with Figure 6, the significant grain growing and coarsening of the striped α_s_ phases is observed in the microstructure of the joints after two-stage annealing. Then, the strengthening effect is canceled by the α phases, which are precipitated in the residual β matrix. Therefore, the strength and plasticity of the joints are reduced [21,22].

The XRD results of Ti600/TC18 joints after stress-relief annealing and two-stage annealing are shown in Figure 7. After annealing, the phases are mainly composed of TiAl, Ti_2_Al, Ti_3_Al, Ti_4_Nb. The morphology, distribution, size, and content of each phase of Ti600/TC18 joints are all affected by heat treatment [23,24].

### 3.2. Tensile Properties of Ti600/TC18 IFW Joints after Heat Treatment

The tensile properties of the Ti600/TC18 joints after stress-relief annealing are shown in Figure 8. It can be seen that the strength of the joints increased firstly and then decreased with the temperature increasing (I-1, I-2, I-3), and they are relatively better at 650 °C (I-2). The tensile strength and yield strength of the joints at room temperature are 1067 MPa and 1022 MPa, respectively, the strength of the joints is at least 1.05 times higher than that of the Ti600 alloy; at 710 MPa and 627 MPa at a high temperature, respectively, the strength of the joints is 1.04 times higher than that of the Ti600 alloy.

There has been a slight improvement in the plasticity of the joints with the temperature increasing. The elongation of the joints at room temperature is all 5%; it is 7.5%, 6%, and 9.5% at high temperature, respectively. In addition, the reduction in the area rate of the joints at room temperature is, respectively, 21%, 20%, and 16%; it is 36%, 30%, and 28% at high temperature, respectively.

According to Figure 4 and Figure 5, it can be seen that after stress-relief annealing, the number of lamellar α′ phases and equiaxed α_p_ phase in the WZ of the joints is more, and they are more densely distributed. Therefore, it has an obvious effect on improving the strength of the joints.

Therefore, the matching relationship between strength and plasticity for Ti600/TC18 joints is better at 650 °C, and the tensile properties of joints are improved.

Figure 9 shows the tensile properties of the Ti600/TC18 joints after two-stage annealing. With the temperature of the high-temperature stage increasing (II-1, II-2, II-4), the strength of the joints at room temperature is increased; it is increased and then decreased at high temperature. The strength of the joints is relatively higher at 880 °C (II-4). The tensile strength and yield strength of the joints at room temperature are 1007 MPa and 972 MPa, respectively; the strength of the joints is at least 0.99 times higher than that of the Ti600 alloy; they are 661 MPa and 601 MPa at high temperature, respectively, and the strength of the joints is at least 0.97 times higher than that of the Ti600 alloy. There is no obvious change in the plasticity of the joints. The elongation of the joints at room temperature is 5%, 4.5%, and 6%, respectively; at high temperature, it is 5%, 6%, and 4.5%, respectively. The reduction in the area rate of the joints at room temperature is 14%, 8%, and 13%; it is 17%, 16%, and 17% at high temperature, respectively.

With the increase in low-temperature stage temperature (II-3, II-4), only the strength of the joints at room temperature is increased slightly. The yield strength of the joints at room temperature is 0.99 times higher than that of the Ti600 alloy at 610 °C (II-4). In addition, the elongation of the joints at room temperature is 6.5% and 6%, respectively, and it is all 4.5% at high temperature. The reduction in the area rate of the joints at room temperature is 14%; it is 13%, 23%, and 17% at high temperature, respectively.

According to Figure 5 and Figure 6, it can be seen that, compared with stress-relief annealing, with the increase in annealing times, the original β grains in the microstructure increased; the size of α_p_ phases increased gradually. The microstructure of two-stage annealing is coarsened. In addition, the morphology and size of the α_s_ phase precipitated in the β phase also changed significantly; the coarsened α_s_ phases are hindered by the intergranular slip, which is unfavorable to the strength and plasticity of the joints.

Therefore, when the two-stage annealing is 880 °C and 610 °C, the matching relationship between the strength and plasticity of the Ti600/TC18 joints is better, and the tensile properties are higher.

The tensile properties of the Ti600/TC18 joints after different heat treatments are shown in Figure 10 and Figure 11. It can be seen that the tensile properties of Ti600/TC18 joints after stress-relief annealing (I) are superior compared to those of the joints after two-stage annealing (II).

The yield strength values of joints can be up to 1.21 times and 1.15 times than that of the Ti600 alloy at room temperature, respectively. They can be up to 1.11 times and 1.10 times than that of the Ti600 alloy at high temperature, respectively. There is no obvious change in the elongation of the joints. In addition, the highest reduction in the area rate of the joints at room temperature and high temperature is, respectively, 21% and 36% after stress-relief annealing; however, they are, respectively, 14% and 23% after two-stage annealing. This is related to the fact that the secondary annealing in the two-stage annealing will cause the α_s_ phases to grow and coarsen; the strengthening effect of lamellar α phases in the β matrix is reduced.

### 3.3. Microhardness of Ti600/TC18 IFW Joints after Heat Treatment

The microhardness values of the Ti600/TC18 joints under different heat treatments are shown in Figure 12. It can be seen that the variation in microhardness values is similar to a “mountain” profile. It shows that the value of the WZ is the highest; it is slightly lower on the Ti600 alloy side, and the value on the TC18 alloy side is the lowest. The reason is that the metastable β grains undergo martensite transformation in the WZ during welding, and the generation of pin-like α phases will improve the strength of the joints. Moreover, XRD results show that the alloy elements are diffused sufficiently and distributed evenly; strengthening phases are formed in the WZ of the joints. However, the welding force on both sides is small, and there is not enough time for dynamic recrystallization, so the microhardness values of both sides are lower.

As shown in Figure 12a, with increasing stress-relief annealing temperature, the microhardness values of the WZ and both sides are increased and then decreased. The microhardness value of the WZ is the highest (378 HV_0.98_) at 650 °C (I-2), followed by that of the Ti600 alloy side (354 HV_0.98_) and TC18 alloy side (358 HV_0.98_). It is relevant that after stress-relief annealing, more metastable β grains are decomposed in a large number of secondary acicular α phases; there are more noticeable effects on improving the strength of the joints. Moreover, the alloy elements diffuse sufficiently and distribute evenly. Therefore, the stress concentration is prevented, and the mechanical properties are improved [25].

As shown in Figure 12b, with the temperature of the high-temperature stage increasing (II-1, II-2, II-4), there is no obvious improvement in the microhardness values of the joints. The microhardness value of the WZ is the highest (370 HV_0.98_), followed by that of the Ti600 alloy side (348 HV_0.98_) and TC18 alloy side (350 HV_0.98_) at 880 °C. As the shapes of αs phases are gradually formed into a basketweave microstructure, it is advantageous to the strength and plasticity. There is no significant change in the microhardness values of the joints with the increase at the low-temperature stage (II-3, II-4).

It can be seen from Figure 13 that the microhardness values are higher after stress-relief annealing compared to that of two-stage annealing. It is related to the fact that the secondary annealing in the two-stage annealing will cause the α_s_ phases to grow and be coarse; it hinders the strengthening effect of lamellar α phases in the β matrix. Therefore, it is adverse to the properties of the joints. However, after stress-relief annealing, the fine recrystallized grains and acicular α phase in the WZ of the joints can significantly improve the performance of the joints.

### 3.4. Fracture Morphology of Ti600/TC18 IFW Joints after Heat Treatment

The macroscopic and microscopic fracture morphologies of the Ti600/TC18 joints after the stress-relief annealing temperature (I-2) and two-stage annealing (II-4) are shown in Figure 14 and Figure 15, respectively. In addition, the tensile fracture occurred at the TC18 side, near the weld. It can be seen that the macroscopic fractures are similar: they have the typical shape of a ductile fracture.

After stress-relief annealing (Figure 14), the microscopic fracture surfaces are relatively rough and consist of micro-pits (position A) and dimples (position B). Furthermore, the dimples are larger and deeper, which indicates that the joint has better plasticity at 650 °C.

The dimples are also seen in the microscopic fractures after two-stage annealing (Figure 15), but they are fewer in number and shallower in depth. Therefore, it is indicated that the plasticity of the joints is lower than that after stress-relief annealing.

The reason for the above phenomenon is that after stress-relief annealing, there are a large number of equiaxed α_p_ phases in the WZ of the joints; they are the core sources of the micropore holes, which are formed at the interface of α/β-phases. After that, as the holes are aggregated and increase in size, the dimples become more and more significant. However, the content and size of the equiaxed α phases are reduced after two-stage annealing; the α/α phase interfaces, α/β phase interfaces, and β/β phase interfaces also diminish correspondingly. The reduction in the above area leads to the grain boundary resistance being reduced during tensile fracture. Therefore, the dimples are fewer in number and shallower in depth [14]. Finally, the joints show poor plasticity, which is consistent with the tensile properties shown in Figure 10 and Figure 11.

## 4. Conclusions

Heat-treatment process conditions are varied to examine the relationship between process parameters, microstructure, and the properties of the Ti600/TC18 IFW joints. The main conclusions are summarized as follows:(1)After the stress-relief annealing, there are α_p_ phases, martensite α′ phases, and residual β phases in the WZ. As the temperature increased, the microstructure of the joints became uniform. Moreover, the strength of the joints increased firstly and then decreased, and there was no significant change in the plasticity of the joints. The tensile properties are relatively better when the temperature is 650 °C.(2)After the two-stage annealing, there are many refined grains, martensite α′ phases, and α_s_ phases with different shapes and sizes in the WZ. With the increase in temperature, the deformed structure α_s_ became gradually spheroid. The strength of the joints is gradually increased, and there was no significant change in the plasticity of the joints. The tensile properties are relatively better when the annealing temperature is at 880 °C and 610 °C.(3)Compared with two-stage annealing, stress-relief annealing has a better strengthening effect on the properties of the joints. It is revealed that the stress-relief annealing microstructure is composed of α′ and β phases, which are beneficial to the properties of the joints. However, the α_s_ phases are coarsened after two-stage annealing, so the tensile properties and the microhardness values of the joints are reduced.

## Figures and Tables

**Figure 1 materials-16-00392-f001:**
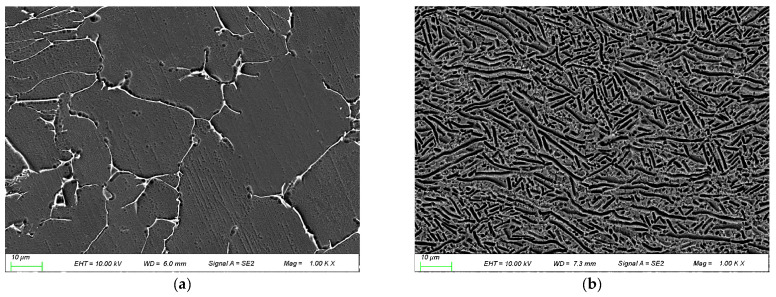
Microstructure of two alloys: (**a**) Ti600 alloy; (**b**) TC18 alloy.

**Figure 2 materials-16-00392-f002:**
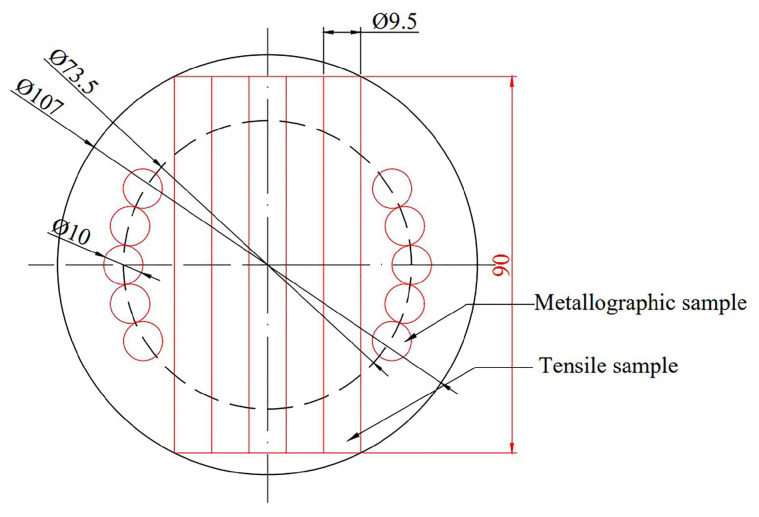
The schematic of the metallographic specimen and tensile specimen sampling.

**Figure 3 materials-16-00392-f003:**
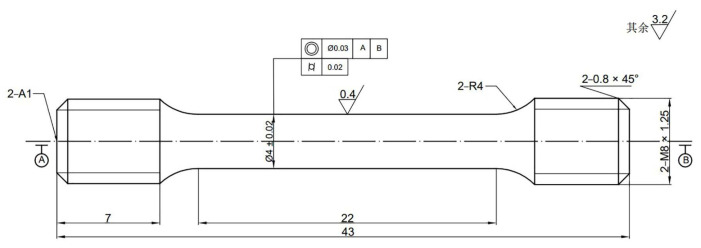
The size of the tensile specimen. 其余 3.2 = the rest 3.2.

**Figure 4 materials-16-00392-f004:**
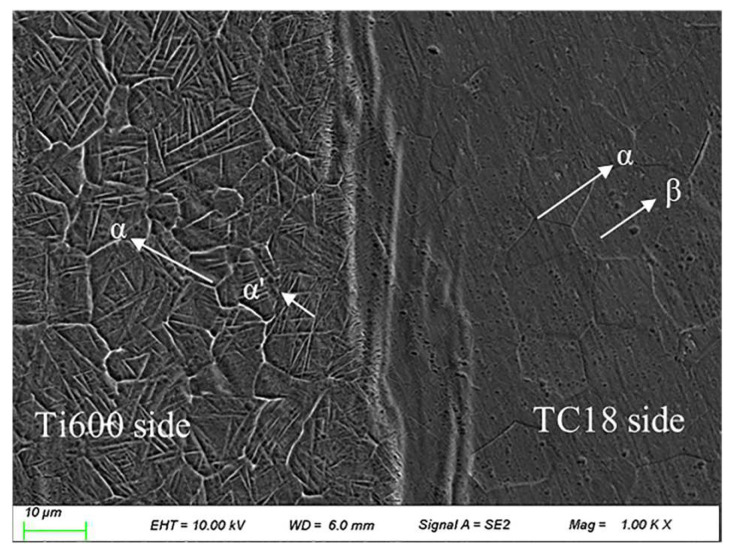
Microstructure of Ti600/TC18 weld before heat treatment.

**Figure 5 materials-16-00392-f005:**
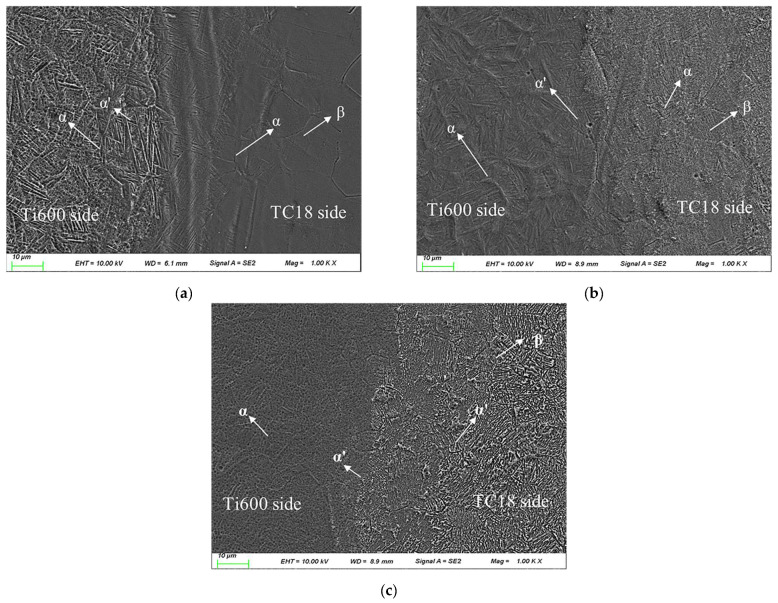
Microstructure of Ti600/TC18 weld after stress relief annealing: (**a**) I-1; (**b**) I-2; (**c**) I-3.

**Figure 6 materials-16-00392-f006:**
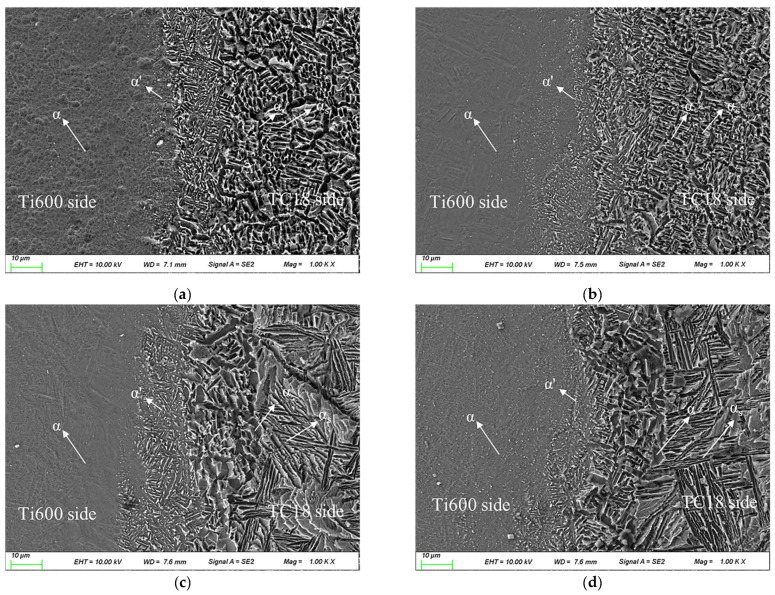
Microstructure of Ti600/TC18 weld after two-stage annealing: (**a**) II-1; (**b**) II-2; (**c**) II-3; (**d**) II-4.

**Figure 7 materials-16-00392-f007:**
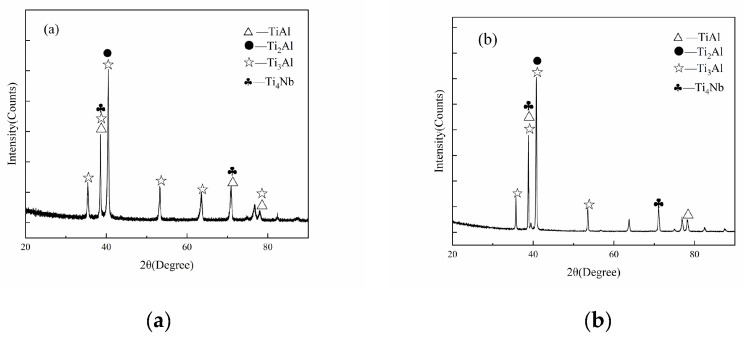
XRD of welds of Ti600/TC18 specimens after heat treatment: (**a**) I-2; (**b**) II-4.

**Figure 8 materials-16-00392-f008:**
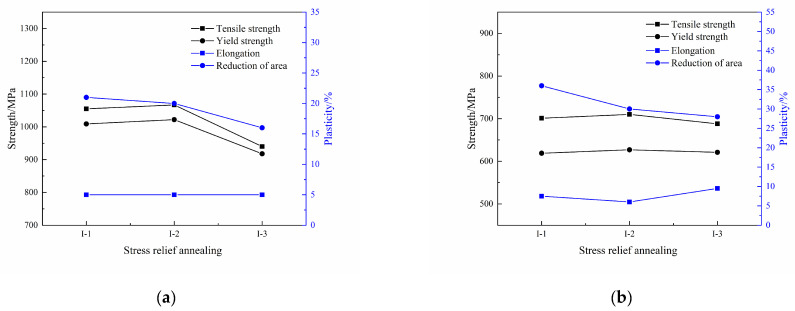
Tensile properties of Ti600/TC18 joints after stress-relief annealing: (**a**) room temperature; (**b**) high temperature.

**Figure 9 materials-16-00392-f009:**
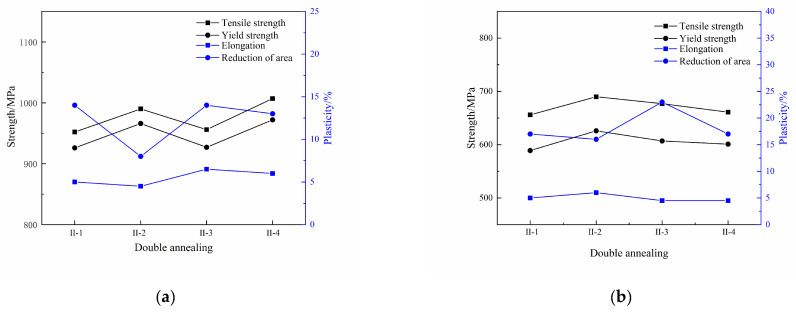
Tensile properties of Ti600/TC18 welded interface after two-stage annealing: (**a**) room temperature; (**b**) high temperature.

**Figure 10 materials-16-00392-f010:**
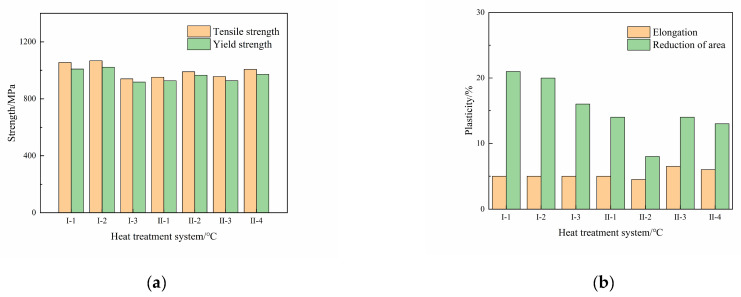
The tensile properties at room temperature of Ti600/TC18 joints after different heat treatments: (**a**) strength; (**b**) plasticity.

**Figure 11 materials-16-00392-f011:**
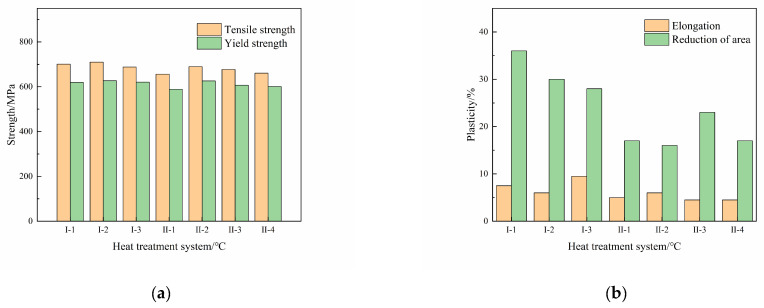
The tensile properties at high temperature of Ti600/TC18 joints after different heat treatments: (**a**) strength; (**b**) plasticity.

**Figure 12 materials-16-00392-f012:**
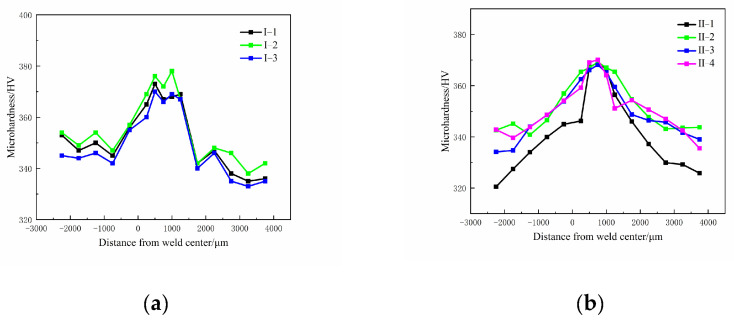
Microhardness of Ti600/TC18 welded joint after heat treatment: (**a**) stress-relief annealing; (**b**) two-stage annealing.

**Figure 13 materials-16-00392-f013:**
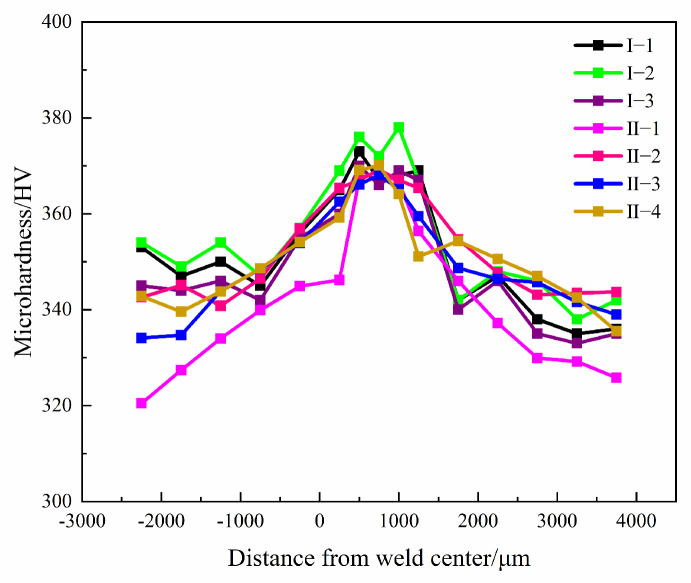
Microhardness of Ti600/TC18 welded joint after different heat treatments.

**Figure 14 materials-16-00392-f014:**
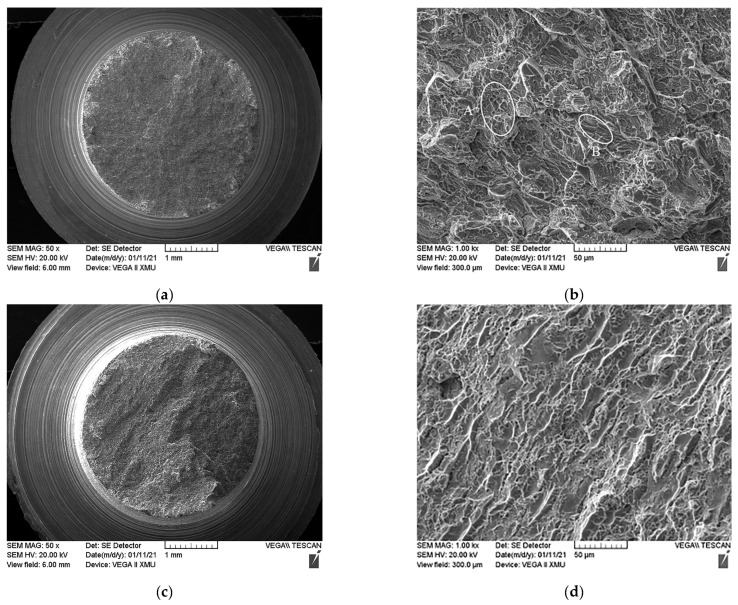
Tensile fracture morphology of I-2: (**a**) macroscopic morphology at room temperature; (**b**) microscopic morphology at room temperature; (**c**) macroscopic morphology at high temperature; (**d**) microscopic morphology at high temperature.

**Figure 15 materials-16-00392-f015:**
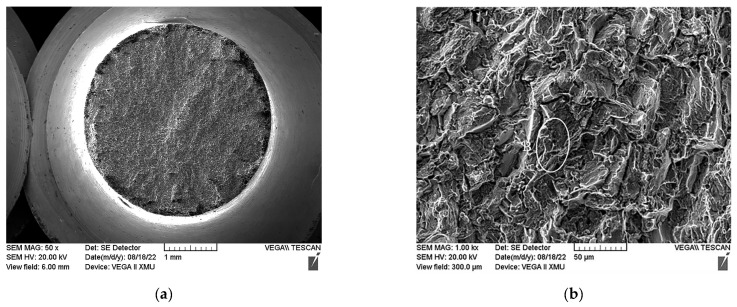
Tensile fracture morphology of II-4: (**a**) macroscopic morphology at room temperature; (**b**) microscopic morphology at room temperature; (**c**) macroscopic morphology at high temperature; (**d**) microscopic morphology at high temperature.

**Table 1 materials-16-00392-t001:** Chemical composition of two alloys (wt.%).

Alloys	Al	Mo	V	Si	Sn	Cr	Nb	Fe	Zr	Ti
Ti600	4.05	0.23	-	0.30	3.87	-	0.06	-	4.49	Bal.
TC18	3.44	4.11	5.49	-	-	0.38	-	1.14	-	Bal.

**Table 2 materials-16-00392-t002:** Mechanical properties of two alloys.

Alloys	Temperature T/°C	Tensile Strength R_m_/MPa	Yield Strength R_p0.2_/MPa	Elongation A/%	Reduction in Area Z/%
Ti600	Room temperature	1021	841	8.5	22
	450	682	564	15.0	32
TC18	Room temperature	1046	995	10.5	38
	450	756	678	16.0	77

**Table 3 materials-16-00392-t003:** Stress-relief annealing heat treatment.

Number	Processing Parameter
I-1	600 °C × 3 h, AC
I-2	650 °C × 3 h, AC
I-3	680 °C × 3 h, AC

**Table 4 materials-16-00392-t004:** Two-stage annealing heat treatment.

Number	High-Temperature Stage	Medium-Temperature Stage	Low-Temperature Stage
II-1	840 °C × 2 h, FC	700 °C × 2 h, AC	610 °C × 6 h, AC
II-2	860 °C × 2 h, FC		610 °C × 6 h, AC
II-3	880 °C × 2 h, FC		590 °C × 6 h, AC
II-4	880 °C × 2 h, FC		610 °C × 6 h, AC

## Data Availability

The data that supports the findings in this study are available from the corresponding author upon reasonable request.

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
