# Peer review of "Effect of Heat Treatment on the Microstructure and Properties of Ti600/TC18 Joints by Inertia Friction Welding"

_materials, 2022, doi:10.3390/ma16010392_

Round 1
Reviewer 1 Report
The manuscript titled "Effect of Heat Treatment on the Microstructure and Properties of Ti600/TC18 Joints by Inertia Friction Welding" is a decent attempt of work.
However, the clarity of the presentation is not significant enough. The authors need to improve the quality of presentation. Also the authors must address the below queries:
1. What is the basis for the choice of heat treatment parameters?
2. Fig.2 represents the schematic sample extraction of IFW samples. However, during friction welding, it is very clear that there is a huge variation from peripheral bonding to bonding of central parts. The authors need to explain the level of quality of the samples.
3. In Fig.4 the interfacial features in the microstructure are not clear.
4. From SEM images, the alpha and beta phase identifications are not very clear.
5. For a clear and specific identification of morphologies of alpha, beta and alpha prime phases, it needs a relatively high magnification images.
6. It is advised to superimpose the (a) and (b) images in Figs 8 and 9.
7. It is mandatory to project few stress-strain diagrams.
8. The data project in tensile data needs a statistical error bar such as in Figs.10 and 11.
9. There are some anomalous in the presented data, such as in Fig.12(a), it shows a medium temperature treatment is showing higher hardness than the other two. Needs detailed and specific comments and analysis on that kind of anomalous presentation.
10. Very important aspect is that the authors must represent the failure location in each tensile tested sample.
11. It is very surprising that the authors could get references related to this work from a specific geographical location only. Why?
Reviewer 2 Report
Manuscript ID Materials-2118195 entitled "Effect of Heat Treatment on the Microstructure and Properties of Ti600/TC18 Joints by Inertia Friction Welding" for journal of Materials has been reviewed.
- The manuscript was interesting and well-motivated. The following list of comments will help to further improve the manuscript:
+1- The “abstract” section should be revised.
+2- The novelty of the study should be further explained (in introduction…).
+3- More references (different studies) should be added to the introduction. (recent studies, 2021-2022)
+4- Tables 1, 2 and 4 should be checked. (Especially numeric values)
+5- Figure 1, 2, 4, 5, 6, 7, 8 and 9, the resolution should be increased. (and magnify)
+6- “In Materials and Methods” …. detail the processes further. (humidity, ambient temperature (appr. value), tester photo, crosshead speed, etc.)
+7- Evaluation of SEM images should be increased.
+8- ….. On the TC18 alloy side, with the temperature of the high-temperature stage increasing (II-1, II-2, II-4), the grain size of αp phases is increased, and its percentage content is slightly decreased. And the αs phases are precipitated from the transformed β matrix and grown progressively [11]. In Figure 6a, there are many fine αp phases distributed on the transformed β matrix, and as phases are began to precipitate diffusely at 840°C. In Figure 6b, the grain size of αp phases is increased, and its percentage content is slowly decreased at 860°C. Moreover, the αs phases are coarsened and have a specific directionality. In Figure 6c, the αp phases are almost disappeared at 880°C; the net-basket microstructure is 160 characteristic of the long and straight lamellar αs phases, which have a particular directionality.………. More explain, Why?
+9- ….. With the increase of low-temperature stage temperature (II-3, II-4), only the strength of the joints at room temperature is increased slightly. The yield strength of the joints at room temperature is 0.99 times higher than that of the Ti600 alloy at 610°C (II-4). And the elongation of the joints at room temperature is 6.5% and 6%, respectively, and it is all 4.5% 220 at high temperature. The reduction of the area rate of the joints at room temperature is 14%; it is 13%, 23%, and 17% at high temperature, respectively…….. This section should be more detailed. (Specific explanations required)
+10- Conclusions section should be enriched.
+11- More literature studies should be added to the introduction and other sections (DOIs given below).
DOI-1 https://doi.org/10.1007/s13369-021-06243-w (about different studies)
DOI-2 https://doi.org/10.35193/bseufbd.1075980 (about different studies)
DOI-3 https://doi.org/10.26701/ems.989945 (about different studies)
----------------------------------------------------
* It will be ready for publication after the specified corrections.
** I want to see article after the revision.
-----------------------------------------------------
Congratulations to the authors.
I wish the authors success in their future academic studies.
Kind regards.
Reviewer 3 Report
I can recommend the publication of this manuscript after a minor revision.
1. Write keywords in alphabetical order.
2. Insert all the SEM parameters, such as magnification, acceleration voltage, working distance, and image pixel resolution (figs. 1, 4, 5, and 6).
3. Insert more details about the statistical method applied and the corresponding software.
4. Lines 81-82: use SI units.
5. Line 90: use a bigger text font in fig. 2.
6. Line 183: Improve the resolution and text font size for fig. 7.
7. Line 203: Improve the resolution and text font size for fig. 8.
8. Line 232: Improve the resolution and text font size for fig. 9.
9. Line 248: Improve the resolution and text font size for fig. 11.
10. Line 276: Improve the resolution and text font size for fig. 12.
11. Line 286: Improve the resolution and text font size for fig. 13.
12. Specify the limits of this study.
13. Explain with more details sentences from lines 150-151, 161-162, 171-172, 225-228, and 283-285.
14. Line 385: minor mistake for ref. [19].
15. References are not written according to the Guide of authors (i.e. sometimes are used: italics for Journals (ref. 12, 13, 14, ...), and sometimes are not used (ref. 17, 18)...... and so on)
16. If possible, authors may consider citing the following references:
1. https://doi.org/10.1007/s11665-020-05270-2
2. https://doi.org/10.1016/j.msea.2019.01.099
3. Ş. Ţălu, Micro and nanoscale characterization of three-dimensional surfaces. Basics and applications. Napoca Star Publishing House, Cluj-Napoca, Romania, 2015.
This paper can be published after the mentioned revisions.
Round 2
Reviewer 1 Report
The queries are not been addressed properly.
Reviewer 2 Report
Manuscript ID: materials-2118195 entitled “Effect of Heat Treatment on the Microstructure and Properties of Ti600/TC18 Joints by Inertia Friction Welding” for journal of Materials has been reviewed.
-Abstract clearly presents objects methods and results.
-Scientific methods are adequately used.
-Terminology is adequate.
-Results are clearly presented.
-Conclusions are logically derived from the data presented.
-Keywords are adequate.
-References are appropriate.
--Decision: accept
The authors have revised the manuscript carefully and the revised version could be published in the journal.
-----------------------------------------------------
Congratulations to the authors.
I wish the authors success in their future academic studies.
Kind regards.